# Neonicotinoid Pesticides Affect Developing Neurons in Experimental Mouse Models and in Human Induced Pluripotent Stem Cell (iPSC)-Derived Neural Cultures and Organoids

**DOI:** 10.3390/cells13151295

**Published:** 2024-07-31

**Authors:** Alessandro Mariani, Davide Comolli, Roberto Fanelli, Gianluigi Forloni, Massimiliano De Paola

**Affiliations:** 1Department of Neuroscience, Istituto di Ricerche Farmacologiche Mario Negri IRCCS, 20156 Milan, Italy; mariani.alessandro@gmail.com (A.M.); davide.comolli@marionegri.it (D.C.); gianluigi.forloni@marionegri.it (G.F.); 2Department of Environmental Health Sciences, Istituto di Ricerche Farmacologiche Mario Negri IRCCS, 20156 Milan, Italy; roberto.fanelli@marionegri.it

**Keywords:** developmental neurotoxicity, neonicotinoids, iPSC, organoids, pesticides

## Abstract

Neonicotinoids are synthetic, nicotine-derived insecticides used worldwide to protect crops and domestic animals from pest insects. The reported evidence shows that they are also able to interact with mammalian nicotine receptors (nAChRs), triggering detrimental responses in cultured neurons. Exposure to high neonicotinoid levels during the fetal period induces neurotoxicity in animal models. Considering the persistent exposure to these insecticides and the key role of nAChRs in brain development, their potential neurotoxicity on mammal central nervous system (CNS) needs further investigations. We studied here the neurodevelopmental effects of different generations of neonicotinoids on CNS cells in mouse fetal brain and primary cultures and in neuronal cells and organoids obtained from human induced pluripotent stem cells (iPSC). Neonicotinoids significantly affect neuron viability, with imidacloprid (IMI) inducing relevant alterations in synaptic protein expression, neurofilament structures, and microglia activation in vitro, and in the brain of prenatally exposed mouse fetuses. IMI induces neurotoxic effects also on developing human iPSC-derived neurons and cortical organoids. Collectively, the current findings show that neonicotinoids might induce impairment during neuro/immune-development in mouse and human CNS cells and provide new insights in the characterization of risk for the exposure to this class of pesticides.

## 1. Introduction

The development of the central nervous system (CNS) is a critical time-window extremely vulnerable to environmental chemicals. Even mild toxic interferences may induce permanent functional and behavioral consequences due to the scanty potential of later repair [1,2]. In this regard, the neurodevelopmental disorders (NDs) prevalence is approximately 3–8% in US infants and the etiology for 3% of these disabilities may be causally linked to the exposure to environmental contaminants, while another 25% is estimated to depend on interactions between chemicals and inherited susceptibilities [3,4]. Chromosomal abnormalities are the most common prenatal disorder with a prevalence of 28.6%, while brain damage in postnatal diagnosis (36.7%) [5].

Documented evidence suggests that pesticides and their residues are widespread [6], and pregnant women and children may represent a highly susceptible population to pesticide toxicity [7,8]. According to recent reports, pesticides can cause neurological damage with different mechanisms including oxidative stress, aberrant cell calcium uptake, neuroinflammation, and changes in neurotransmitter levels [9,10]. Although neurodevelopmental neurotoxicity has been demonstrated only for high-level exposure to certain pesticides in utero and in early childhood [11,12], studies on the potential effects of chronic low-dose exposure are nowadays lacking or inadequate for most chemicals in use [13]. Neonicotinoids (neonics) are a class of synthetic, nicotine-derived insecticides that are primarily used worldwide to protect crops from pest insects and domestic animals from fleas [14,15]. Based on a risk assessment of the European Food Safety Authority (EFSA), the European Commission prohibited their use in bee-attractive crops in 2013 [16] and banned all outdoor uses of three substances (clothianidin, imidacloprid, and thiamethoxam) in 2018 [17], in response to an increasingly strong body of research suggesting they are lethal for pollinators such as bees. However, neonic use is not prohibited in many other places in the world, and they are currently under debate in the US market, posing the risk of a widespread ecosystem contamination and a serious threat to food webs [18]. Neonicotinoids have been reported to act as agonists of nicotinic acetylcholine receptors (nAChRs) selectively displacing acetylcholine in insects, leading to depolarization of the cell, and eventually resulting in the death of insects [19,20]. Their selective toxicity to insects has been attributed to a higher binding affinity to the postsynaptic nAChRs in insects compared to vertebrates [14]. Although mammals, including humans, are exposed to neonicotinoid pesticides, very little is known about their effects on the mammalian CNS. The fundamental role of nAChRs in CNS development [21,22] and their widespread presence in non-neuronal cells (e.g., astrocytes, microglia, and endothelial cells) [23,24,25] suggest that mammals could be highly vulnerable to neonicotinoid exposures in the earlier phases of organ development. Experiments in vitro showed that mammalian nAChRs in CNS neurons are quickly targeted by neonicotinoids at concentrations ranging between 1–10 µM [26,27,28]. Acute exposure to the same concentrations is able to change the cell membrane properties and induce significant excitatory calcium influxes in neurons [29,30]. Experiments on human intestinal cells have suggested that they might be absorbed by active transporters in the intestines [31,32]. These in vitro data are supported by different studies in animals showing the ability of neonicotinoids to readily cross the blood–brain barrier [33] and that some metabolites have a higher affinities for mammalian nAChRs, similar to those of nicotine [34]. In utero exposure to neonics has been correlated to neurobehavioral deficits (impairment in sensorimotor performances) associated with increased acetylcholinesterase activity and astrogliosis in offspring rats [35], while oral administrations can also lead to immunosuppression (either directly or by stress mechanisms) and impairment in the neuroendocrine system [36,37,38,39]. This evidence suggests that neonicotinoids may actually affect mammalian nAChRs to a greater extent than previously believed on the basis of binding-assay data, representing a potential risk for neurodevelopment in humans. The effects of insecticides on human health have been recently reviewed, and their possible contribution to autism spectrum disorder in combination with other environmental risk factors has emerged [40,41]. The neurotoxic effects of these compounds on mammals has been suggested including decreased neurogenesis, induced neuroinflammation, and affected neurobehavioral toxicity [42]. However, the adverse effects of these compounds on mammals CNS are not yet fully disclosed, especially for the exposure to low pesticide doses. Through a mass-spectrometry-based integrated approach, we recently showed that imidacloprid (IMI) and its metabolites are able to pass through both the blood–brain barrier and the placenta in mice, thus possibly targeting the nervous cells in adults and fetuses [43].

### Aim of This Study

Here, we investigated the impact of chronic exposure of mammal brain and neural cells to imidacloprid, clothianidin, and dinotefuran which represent some of the actual dominating commercial active substances in their respective chemical sub-group of neonicotinoids. We show that, at doses consistent with human real-life exposure, they can affect neurons and glia development by altering different sensible targets of neurotoxicity (i.e., viability, metabolic markers, synaptic proteins, neurofilament composition, etc.) in in vitro and in vivo mouse models and in human iPSC-derived cells and three-dimensional self-assembling brain organoids (namely, cortical organoids, hCOs). The results collected from these analyses showed an interesting picture of peculiar neurodevelopmental alterations induced by imidacloprid on mammalian CNS cells.

## 2. Materials and Methods

### 2.1. Overall Experimental Design

We report here in Figure 1 the experimental methods and analysis that were developed through the study.

### 2.2. Chemicals

The neonicotinoids tested in this project were imidacloprid, clothianidin, and dinotefuran (Sigma Aldrich, Saint Louis, MO, USA) representing some of the actual dominating commercial active substances in their respective chemical sub-group of neonicotinoids. Nicotine and mecamylamine (Sigma Aldrich, Saint Louis, MO, USA) were used as the reference nAChRs agonist and non-competitive antagonist, respectively.

### 2.3. Animal Care

Procedures involving animals and their care were conducted in conformity with the institutional guidelines at the Institute for Pharmacological Research Mario Negri–IRCCS in compliance with national (dlgs 4 marzo 2014, n. 26) and international laws and policies (European Directive 2010/63). Ethical approval for animal procedures was obtained from the internal Ethical Committee at the Istituto di Ricerche Farmacologiche Mario Negri–IRCCS and from the Italian Ministry of Health (authorization number 154-2016-PR).

### 2.4. Primary Cell Cultures

Primary cultures were obtained from the hippocampus, cerebellum, or cortex of 12-day-old (E12) C57BL/6J mouse embryos as previously reported [44] by adapting different protocols [45,46,47,48,49]. Briefly, tissues were dissected and mechanically disrupted, exposed to DNAse and trypsin (Sigma Aldrich, Saint Louis, MO, USA), and centrifuged through a bovine serum albumin (BSA) cushion. Cells obtained at this step were a mixed neuron/glia population. The following steps were adopted for the different cell cultures.

#### 2.4.1. Hippocampal, Cerebellar, and Cortical Cultures

Cells obtained from the hippocampus, cerebellum, or cortex of mouse embryonic brain were plated at a density of 1.5 × 10^5^ or 3 × 10^5^ cells/cm^2^ into 24- or 96-well plates, respectively, previously pre-coated with poly-L-lysine (Sigma Aldrich, Saint Louis, MO, USA) and maintained in neurobasal-based complete culture medium (CCM) containing B27 supplement, brain-derived neurotrophic factor (BDNF, Miltenyi Biotec, #130-096-286, Bergisch Gladbach, Germany) and horse serum (for complete media composition please refer to Appendix A). To obtain neuron-enriched cultures, after 48 h from plating, cells were treated with 5 µM cytosine arabinoside (Ara-C, Sigma Aldrich, #C6645) for 24 h to inhibit the proliferation of glial cells and were subsequently maintained in serumless medium. Glutamate (25 µM, Sigma Aldrich, #G1501, Saint Louis, MO, USA) was added to the cultures for the first 4 days in vitro (DIV).

#### 2.4.2. Glial Cultures

Glia-enriched cultures were obtained from the cerebral cortices by plating the different cell populations at a density of 40,000 cells/cm^2^ into flasks (pre-coated with poly-L-lysine). When maintained in DMEM-based astrocyte medium (DAM), neurons die within 2 weeks due to the lack of neurotrophic factors. Purified microglia were obtained by shaking confluent mixed glial cultures overnight at 275 rpm, as previously reported [44]. Floating cells were collected and seeded at a density of 200,000 cells/cm^2^ into 96-well plates for neurotoxicity analysis. Astrocyte-enriched cultures were obtained by treating the glial cultures, from which microglia had been previously harvested, with 60 mM L-leucine methyl ester (to selectively kill microglia; Sigma Aldrich, Saint Louis, MO, USA) for 90 min. For the toxicity analysis, astrocytes were seeded at a density of 200,000 cells/cm^2^ into 96-well plates.

#### 2.4.3. “Sandwich” Neuron/Glia Cocultures

To evaluate the in vitro neuron developmental alterations following neonicotinoid exposure, neuron/glia “sandwich” cocultures were established. In this model, the two cell types were not in contact but shared a common medium, and a low-density neuron-enriched cell population with healthy morphology and good differentiation could be obtained [44,50,51]. Briefly, the hippocampal cell suspension obtained from the dissociation of embryo brain tissues was plated on glass coverslips with paraffin dots and maintained in serumless medium (CCM without serum). On the 2nd DIV, coverslips with cultured neuron were transferred into 12-well plates containing the astrocyte feeder layer. The coverslips were inverted so the paraffin dots created a narrow gap that prevented the contact of the two cell types; however, they shared a common medium. The day before treatment, microglia were added to the cultures (10% of the astrocytes number), and cocultures were maintained hereinafter in CCM. Media compositions are reported in Appendix A.

### 2.5. iPSC-Derived Cell Cultures and Cortical Organoids

A commercial human iPSC line obtained from healthy donors was used in this study (Episomal Human iPSC Line, ref A18943PIS, Lot V2.0). The iPSC line was cultured and expanded in feeder-free conditions by passaging every 5–7 days when they reached 70−80% confluence in a serum-free, stabilized cell culture medium (mTeSR™ Plus, Stemcell Technologies Inc., #100-0276, Vancouver, BC, Canada). To obtain differentiated neural cultures, neural progenitor cells (NPCs) were derived from iPSC by chemically blocking TGF-β/BMP-dependent SMAD signaling (STEMdiff™ SMADi Neural Induction Kit, Stemcell Technologies, #08581, Vancouver, BC, Canada) with a monolayer culture protocol [52]. NPCs were then expanded when confluent through different passages (NPCs from the 6th to the 15th passage, P6–15, were used here) before differentiation into monolayer neuron or astrocyte cultures, or generation of 3D self-assembling cortical organoids (hCOs), according to published protocols [53,54,55,56]. Briefly, to obtain differentiated neurons, NPCs were dissociated with Versene Solution (Gibco ^TM^, Scotland, UK) and plated into poly-ornithine/laminin coated plates at a density of 5 × 10^4^ cells/cm^2^, and cultured with a neuron differentiation medium containing specific neurotrophic factors (NDM, see Appendix A, for details). For astrocyte cultures, dissociated NPCs were plated onto Geltrex (Gibco^TM^)-coated 24-well plates at a density of 5 × 10^4^ cells/cm^2^ in an astrocyte differentiation medium (ADM, Appendix A) for 28 days. hCOs were generated as previously described [56] with minor modifications. Briefly, at 90% confluence, NPCs were detached with StemPro™ Accutase™ Cell Dissociation Reagent (Gibco^TM^), and 10^5^ cells were plated in low-attachment 96-well plates. Cell aggregates were grown in hCO differentiation medium (Appendix A) at 37 °C in an atmosphere of 5% CO_2_, under constant gyratory shaking for up to 8 weeks.

### 2.6. Immunocytochemistry

After treatment, primary mouse and human iPSC-derived monolayer cultures were fixed with 4% formaldehyde and, when required, permeabilized with 0.2% Triton X-100 (Sigma Aldrich) solution in PBS. Staining was carried out by overnight incubation with the primary antibody in blocking solution (10% FBS in PBS). Primary antibodies were as follows: neurofilament 200 (NF200, 1:500; Sigma Aldrich, Saint Louis, MO, USA, #N4142), α7-nAChR (1:250; Santa Cruz Biotechnology, Heidelberg Germany, #sc-1447), GFAP (1:1000; Merck Millipore, Temecula, CA, #MAB3402), CD11b (1:500; eBioscience, Life Technologies, Bleiswijk Netherlands, #14-0112), S100β (rabbit, 1:500; Biorbyt, UK, #orb388641), anti-doublecortin (DCX, 1:250; Santa Cruz Biotechnology, Heidelberg, Germany #sc-271390), and synaptophysin (SYN, 1:250; Sigma Aldrich, Saint Louis, MO, USA, #S5768). Cell nuclei were labeled with Hoechst 33258 (Sigma Aldrich, Saint Louis, MO, USA, #14533) via incubation with a 250 ng/mL solution. Appropriate fluorescent secondary antibodies conjugated to different fluorochromes (Dy-light; Rockland Immunochemicals, Limerick, PA, USA) were used at 1:1000 dilution. When needed, double or triple staining was performed by overnight incubation of the cultures with each primary antibody separately. For fluorescent signal quantification, Z-stack pictures of stained cells were acquired at 600× or 1200× magnification with a laser scanning microscope (Olympus Fluoview BX61 with a FV500 confocal system). Deconvolution, 3D reconstructions, and data analysis were performed with the Imaris software version 7.2 (Bitplane Inc., Oxford Instruments, Oxfordshire, UK). For neurogenesis analysis, data were expressed as the volume of doublecortin (DCX) (a cytoskeleton-associated protein expressed by immature neurons) normalized for the total volume of neurofilaments. For synaptic analysis, data were calculated as the number of SYN-positive puncta on the NF200 volume and expressed as percentage of the control. The complexity of dendritic arborization was evaluated by an automated Sholl analysis on the NF200 signal, and the number of dendritic branches was calculated for each recognized neuron.

Whole-mounting hCO staining was performed, as follows. hCOs were fixed with 4% formaldehyde for 1 h at room temperature, followed by dehydration with different methanol concentrations. Permeabilization was then achieved using 3% BSA, 0.5% Triton X-100 in PBS for 2 h. hCOs were then incubated with the primary antibodies GFAP (mouse; Millipore, Temecula, CA, USA #MAB3402) and MAP-2 (rabbit; Merk Life Science Srl, Darmstadt, Germany, #AB5622) diluted 1:200 in blocking solution (5%BSA, 0.1% Triton X-100 in PBS) for 24h in agitation (50 rpm) at 37 °C. Incubation with the secondary antibodies anti-mouse 488 and anti-rabbit 647 (Rockland Immunochemicals, Limerick, Pennsylvania, USA) diluted 1:250 in blocking solution was then performed for a further 24 h in agitation at 37 °C. An additional 2 h incubation with Hoechst 33258 (Sigma Aldrich, Saint Louis, MO, USA, #14533) diluted 1:200 in PBS to identify cell nuclei followed when needed. hCOs were then dehydrated sequentially in 50% and 100% methanol solutions and exposed to Visikol HISTO-M tm (Sigma Aldrich, Saint Louis, MO, USA) solution for 30 min in Ibidi, μ-Slide 2 Well Co-Culture plates (Ibidi GMBH, Gräfelfing, Germany) to achieve tissue clearing before image acquisition. Confocal microscopy was performed on a Nikon A1 confocal scan unit, managed by NIS elements software version 4.51. Large field acquisition was obtained with a 20 × 0.5 NA, using a 10% image overlapping to allow stitching. Each image had a pixel size of 0.6 µm and was acquired over a 200–250 µm z-axis (step size of 2.6 µm).

### 2.7. Cell Viability and Toxicity Assessment

For cell viability assessment in mouse cultures, neurons, microglia, or astrocytes obtained from the hippocampus, cerebellum or cortex of mouse embryos were exposed to the different neonicotinoids (diluted to the appropriate concentrations in culture medium) for 72 h. Cells maintained in culture medium were the control conditions. Cell viability was assessed by the CellTiter96 non-radioactive cell proliferation assay (MTS test; Promega, Madison, WI, USA) following the manufacturer’s instructions. For human iPSC-derived cell cultures and hCOs, chronic treatments with IMI were performed 3 times/week, and 50 µL samples were collected from the medium in the same wells at the indicated time points and frozen in a storage buffer at a 1:10 dilution. Samples were thawed and further diluted 2.5-fold in the storage buffer before measuring lactate dehydrogenase (LDH) levels. A bioluminescent plate-based assay for quantifying LDH release into the culture medium was used (LDH-Glo™ Cytotoxicity Assay, Promega, Madison, WI, USA).

### 2.8. In Vitro Evaluation of Microglia Activation

To assess neonic effects on microglia, the mRNA expression of TNFα and Ym1 was analyzed by real-time reverse transcription PCR (RT-PCR) in microglia cultures pre-treated with IMI for 72 h in resting condition or after the pro-inflammatory stimulus induced by treatment with 1 µg/mL lipopolysaccharide (LPS) for 24 h. Samples were collected after treatments, immediately frozen on dry ice and stored at −80 °C until analysis. Total RNA was extracted with the RNeasy kit (Qiagen). Samples were treated with DNase (Life Technologies, Bleiswijk, Netherlands) and reverse-transcribed with random hexamer primers using Multi-Scribe Reverse Transcriptase (Life Technologies, Bleiswijk, Netherlands). RT-PCR was performed with b-actin as a housekeeping gene. Relative gene expression was determined by the ΔΔC_t_ method using mean C_t_ from three replicates per sample. The data are expressed as the log 2-fold difference over the untreated group. The TNFα concentration in the cell culture supernatant (conditioned media) was quantified by solid-phase sandwich ELISA (eBioscience, Inc., Frankfurt am Main, Germany). The samples from each experiment were tested in triplicate, according to the manufacturer’s instructions.

### 2.9. In Vivo Model of Prenatal Exposure to Imidacloprid or Nicotine

#### 2.9.1. Experimental Groups

Overnight mating was performed with two female and one male C57BL/6N mice (purchased from Charles Rivers-Italy), and the day of sperm detection in the vaginal smear was considered as GD 0. Dams were then divided into different experimental groups and treated once a day from GD 6 to GD 9 with imidacloprid by gavage or nicotine by intranasal injections:Control (sterile water)Imidacloprid, human exposure (H. Exp., 118 µg/kg bodyweight/inj)Imidacloprid, acceptable daily intake (aDi, 4.1 mg/kg bodyweight/inj)Imidacloprid, 10× ADI (41 mg/kg bodyweight/inj)Nicotine, low dose (26.5 µg/kg bodyweight/inj)Nicotine, high dose (53 µg/kg bodyweight/inj)

For each experimental group, brains were explanted from newborn mice of at least three litters for ex vivo analysis.

#### 2.9.2. Ex Vivo Analysis

The expression of synaptophysin, doublecortin (DCX), and CD11b was analyzed in the brain explanted from imidacloprid/nicotine prenatally exposed newborn mice on serial paraffinized brain slices via immunohistochemistry.

#### 2.9.3. Immunohistochemistry

Dissected tissues were fixed with 10% formalin for no less than 48 h at room temperature. After removal of formaldehyde by running tap water, tissues were dehydrated in EtOH baths with increasing concentrations from 30% to 100%. Tissues were then cleared in xylene for 1h and incubated in a 65 °C paraffin bath and kept overnight at 60 °C. The paraffin-embedded tissue blocks were then sectioned in 20 μm thickness slides on a microtome (LEICA RM2125RT, Leica Biosystems) floating in a 37 °C water bath containing deionized water. Slides were allowed to dry overnight and stored at room temperature until ready for use. Serial paraffinized 20 µm thick sagittal brain slices were collected, and the expression of the above-mentioned markers were detected in hippocampi and cerebella of postnatal day 0 (PND0) mice, as follows. Brain slices put on poly-L-lysine-coated glass slides (3/slides) were first deparaffinized and rehydrated with xylene and EtOH baths with decreasing concentrations from 100% to 30%, respectively. Subsequently, antigen retrieval in pH 6 citrate buffer was followed by 30 min incubation with 10% fetal bovine serum in 1% Triton X-100 in Tris–HCl-buffered saline (TBS, pH 7.4). Sections were incubated first with the primary antibody against synaptophysin (1:200, Sigma Aldrich), DCX (1:200; Santa Cruz Biotechnology, Heidelberg Germany), or IBA-1 (rabbit, 1:5000; Wako Chemicals, Neuss Deutschland, #019-19741) in a mixed solution containing 10% fetal bovine serum/1% Triton X-100/TBS for 24 h, at 4 °C, then with the biotinylated (for synaptophysin and Iba-1) or fluorescent (for DCX) secondary antibody (1:200, Vector Labs, Newark, CA, USA). Pictures of stained slices were acquired at 200X magnification with a laser scanning confocal microscope (Olympus Fluoview BX61 with a FV500 confocal system). The data analyses (signal intensity, signal area, or number of positive cells) were performed using Imaris version 7.2 (Bitplane Inc., Oxford Instruments, Oxfordshire, UK) or ImageJ software version 1.43 (National Institutes of Health, https://imagej.net/nih-image/ accessed on 1 January 2024), and the raw data were normalized to mean levels in untreated brains of newborn mice (control group).

#### 2.9.4. Statistical Analysis

The data were analyzed via one-way ANOVA and Dunnett’s or Tukey’s test or via two-way ANOVA and Dunnett’s or Bonferroni’s post-test, using GraphPad Prism version 6.01 (GraphPad Software Inc., Boston, MA, USA). The limit of statistical significance was set at *p* < 0.05. The LC50 (lethal concentration 50, concentration inducing 50% cell mortality), LOAEL (lowest observed adverse effect level) and NOAEL (no observed adverse effect level) were calculated using GraphPad Prism software 6.01 (logarithmic transformation of X-values and non-linear regression, sigmoidal dose–response analysis with variable slope, with bottom and top constrains set at 0 and 100, respectively). Values are given ± 95% confidence intervals.

## 3. Results

### 3.1. Estimated Neonicotinoid Experimental Doses

Neonicotinoid concentrations for in vitro treatments were calculated from the estimated diet intake in women of childbearing age (13–49 years) [57,58,59] or the acceptable daily intakes (ADI) set by the governmental and regulatory agencies [60,61,62]. These values were then adjusted for the reported linear association between oral intake (mg/kg bw) and plasma concentrations (µM) of imidacloprid (0.357; Table 1) [60].

It was then assumed that plasma concentrations in pregnant women/mice are comparable to those in umbilical cord blood and in fetal brain, as we recently showed for mouse dams [43]. An estimated real-life (eFBC) and admissible (aFBC) concentration for fetal brain exposure was thus determined for each tested compound (Table 1). Increasing concentrations (up to 10,000-fold aFBC) were also tested. Nicotine was used at the same order of magnitude of the different neonicotinoids. The doses of imidacloprid for in vivo treatment of pregnant mice were calculated from the estimated women’s daily intake during pregnancy (0.470 mg/Kg bw) and the derived exposure daily intake (human exposure dose, H. exp: 1.75 µg/kg bodyweight/day [59]) or the acceptable daily intake (ADI: 60 µg/kg bodyweight/day [60]) and then corrected for the different gestation length in humans or mice and divided for the number of oral administrations scheduled for the animal model of prenatal exposure (Table 1). Nicotine was administered by intranasal injection at doses calculated from the concentration revealed in smokers [63,64,65] and adjusted for our experimental protocol with the following final doses: low dose: 26.5 µg/kg bodyweight/inj, high dose: 53 µg/kg bodyweight/inj.

### 3.2. Mouse Primary Culture Characterization and α7-nAChR Expression

We initially characterized the cellular abundance in our in vitro setting for the neuron cultures obtained from different brain areas (the hippocampus, cerebellum, and cortex) at different time points (6 and 14 days in vitro, DIV) by immunocytochemistry (Appendix A). Specific markers for neurons, astrocytes, or microglia (NF200, GFAP, and IBA-1, respectively) were used, and the number of positive cells for each specific antibody was assessed. Neurons were predominant at 6 DIV (>96.7%, Appendix A), but glia contamination ranging from 0.63% to 2.48% were revealed (Appendix A). Cultures maintained for 14 DIV and treated with cytarabine (Ara-C) and serum deprivation were mainly composed of NF200-positive neurons, since no microglia were detected and astrocytes were reduced to 0.39–0.69% of total cells (Appendix A). The α7-nAChR expression was verified for the different cell populations in mouse primary neuron/glia cocultures by immunocytochemistry with cell-specific markers and antibody for α7-nAChR. Immunopositivity for α7-nAChR in cultured neurons was detected already after 3 DIV and became widespread in hippocampal (Figure 1A–C), cortical (Figure 1D–F) and cerebellar (Figure 1G–I) NF-200-positive neurons after 14 DIV. As for glial cells, CD11b-positive microglia were specifically stained by the α7-nAChR antibody (Figure 1J–L), while no signal was detected in mouse purified astrocyte cultures (Figure 1M–O).

### 3.3. Neonics Induce Neurotoxicity and Affect Microglia Viability and Inflammatory Status but Is Not Effective on Astrocytes in Mouse Primary Cultures

The effects of neonics on the viability of neurons and glial cells were analyzed in primary cultures obtained from the cerebellum, hippocampus, and cortex of 12-day-old mouse embryos. Treatments were performed over 72 h with different concentrations of the pesticides (eFBC, aFBC, 100×, 1000×, or 10,000× aFBC). Cell viability was then assessed by two common cytotoxicity tests: the MTS and LDH release assay. The effects of neonics on early-developing neurons were detected at the 3 DIV. Clothianidin showed higher toxicity compared to imidacloprid and dinotefuran. Significant neurotoxic effects were indeed observed down by nanomolar levels (LOAEL: 270 nM, corresponding to the aFBC) for clothianidin in hippocampal cultures (Figure 2B), while imidacloprid and dinotefuran significantly affect neuron viability only from 170 μM (Figure 2A) and 6.2 mM (Figure 2C), respectively.

When comparing the different brain region origins, hippocampal cultures showed higher sensitivity to imidacloprid and clothianidin than cortical and cerebellar neurons, where the LOAEL increased to 1.7 and 2.7 mM (10,000× aFBC), respectively, suggesting a region-specific neuron susceptibility (Figure 2A,B). Nicotine treatments affected neuron viability with a trend similar to imidacloprid, with significant effects observed from the 1000× aFBC, namely, 500 µM, in hippocampal and cerebellar cultures. Cell viability was also determined in neuron cultures after 16 DIV. In this case, in vitro treatments were performed from 13 to 16 DIV. At this developmental stage, neonicotinoids decreased the neuron viability by lower concentrations than in early-developing neurons (Figure 2E–H). Interestingly, neurons were more sensitive to imidacloprid and dinotefuran, which induced significant effects down by eFBC or aFBC levels (LOAEL 5 and 620 nM, respectively; see Figure 2E,G) in hippocampal and cortical cultures, compared to clothianidin, whose toxic effect was significant from 100× aFBC (Figure 2F). Nicotine induced neurotoxicity with a trend similar to neonics, showing higher effects on hippocampal neurons, where a significant cell death was induced from the eFBC concentration (Figure 2H). We then verified the involvement of nAChRs in neonicotinoid-mediated neurotoxicity via co-treatments with the specific non-competitive antagonist mecamylamine (MECA). To this purpose, a first set of experiments on the toxicological profile of MECA was performed on hippocampal cultures. MECA showed a dose-dependent toxicity on hippocampal neurons, with the highest nontoxic concentration being 50 μM (NOAEL: 50 μM; Figure 3).

Based on these results, hippocampal neurons were subsequently co-treated with different concentrations of MECA (up to 50 μM) and 17 μM imidacloprid or 50 μM nicotine. Mecamylamine prevented neuron death induced by imidacloprid and nicotine in a concentration-dependent manner, with the lowest concentration of mecamylamine able to significantly reduce neonics’ neurotoxic effects by 10 μM (F_interaction_ (2, 102) = 6,23; *p* < 0.01; Figure 4).

Moving to glial cells, we assessed the effects of imidacloprid on the immune responses mediated by microglia. To understand if the pesticide alone was able to mediate activation, purified cultures of microglia were treated with imidacloprid (aFBC or 1000× aFBC) for 72 h. Cells were then analyzed by RT-PCR for pro-inflammatory (M1) or anti-inflammatory (M2) phenotype markers (TNFα and YM1, respectively) up to 1000× aFBC (170 μM). Imidacloprid alone did not induce significant alterations in the expression levels of TNFα and YM1 mRNA up to 1000× aFBC (170 μM; Figure 5A).

We then decided to test if pre-treatment with IMI could affect microglia response to a general pro-inflammatory stimulus. To this aim, 72 h imidacloprid pre-treatment was performed before co-treatment with LPS 1 μg/mL. The pre-treatment with the higher dose (170 μM) of the pesticide reduced the pro-inflammatory responses of immune cells to LPS as highlighted by the reduction (*p* < 0.05 vs. LPS) in the level of TNFα mRNA by about 53% compared to LPS alone (Figure 5B). We then analyzed if this reduction in TNFα mRNA was correlated with an altered production of the cytokine by microglia. The release of TNFα in the conditioned media was thus measured in purified microglia cultures stimulated with LPS and pre-treated with imidacloprid. In this setting, the levels of TNFα released in the conditioned media was not directly affected by imidacloprid alone if compared to untreated conditions (Figure 5C). However, IMI was able to modulate the pro-inflammatory effects exerted by LPS treatment on the TNFα release. Indeed, the pre-exposure with IMI significantly reduced the cytokine release elicited by 1 μg/mL LPS (*p* < 0.001 for 170 μM IMI vs. LPS; Figure 5C). On the other hand, exposure to either the neonicotinoids or nicotine (up to 1.7–6.2 mM) did not affect astrocyte viability in purified cultures, in accordance with the absence of nicotine receptor expression observed in these cultures.

### 3.4. Imidacloprid Induces Neurodevelopmental Alterations in Mouse Neuron Cultures

Neurodevelopmental alterations induced by non-lethal concentrations of the neonicotinoids on developing neurons was performed on a “sandwich” neuron/glia coculture setting. Hippocampal neurons cocultured with glia were treated with imidacloprid on the 10 DIV for 72 h (a period that is reportedly characterized by a relevant increase in the synaptic density). Through immunocytochemistry, the effects of neonicotinoids were analyzed in terms of their neurofilament composition, dendritic arborization, and synaptic marker expression. Imidacloprid (up to 17 µM, corresponding to 100× aFBC) did not significantly affect neuron maturation, expressed as the ratio of immature (DCX) over mature (NF200) neurofilaments (Figure 6A). On the other hand, the dendritic arborization complexity of NF200-positive neurons, expressed as the number of dendritic branches for each neuron, was dose-dependently reduced (Figure 6B). An increase in the density of synaptophysin-positive puncta on neurofilaments was also observed after treatment with the highest imidacloprid concentration (17 µM IMI, *p* < 0.001 compared to the control condition; Figure 6C).

### 3.5. Imidacloprid Induces Toxicity in Human iPSC-Derived Neural Cultures and Brain Organoids

We checked for the expression of α7-nAChR in iPSC-derived neurons (i-neu) and astrocytes (i-astro). NF200-positive i-neu (Figure 7A) differentiated for 28 DIV widely expressed the receptor-specific signal (Figure 7B, merged signals in Figure 7C).

Differently from mouse primary astrocytes, S100β-positive i-astro showed a specific staining for α7-nAChR (Figure 7D–F). I-neu were then chronically exposed to imidacloprid by repetitive treatments with different concentrations (up to 17 µM, corresponding to 100× aFBC; Figure 8A–E) of the pesticide every other day, from 3 to 28 DIV. Cell viability was then analyzed with the MTT assay at 3, 10, 14, 21, and 28 DIV. By this protocol, it was possible to observe a transient toxicity induced by IMI at 10 and 14 DIV (Figure 8B,C). At 14 DIV in particular, a significant decrease in cell viability was observed since the lowest concentration tested (eFBC, *p* < 0.01 vs. CTR). In line with the documented expression of α7-nAChR in i-astro, exposure to IMI induced astrocyte death, as determined by the increased release of LDH in the conditioned media after 72 h treatment (Figure 8D).

We then investigated the effect of chronic exposure to IMI in hCOs using a 3D iPSC-derived model which includes differentiated astrocytes and neurons (Figure 9A). This model supports the adoption of a chronic treatment protocol with 5 nM or 17 μM IMI (eFBC and aFBC, respectively) three times a week from 11 DIV until the complete maturation of the hCOs, i.e., 60 DIV. Conditioned media were collected before the treatment (0 DIV) and at 25, 35 49, and 60 DIV, and LDH release was measured at the different time points. As shown in Figure 9, 17 μM IMI induced cell death at 25 (*p* < 0.01), 35, and 60 DIV (*p* < 0.001 vs. CTR), while no significant effects were revealed for the lower concentration.

### 3.6. Neurodevelopmental Alterations Induced by Prenatal Exposure to Imidacloprid or Nicotine in Mouse Brains

To verify the in vivo effects of imidacloprid and nicotine on neurodevelopment, ex vivo analyses were performed on brains obtained from newborn mice prenatally exposed to imidacloprid or nicotine via dam treatments. The amount of synaptic (synaptophysin) and neurogenesis (DCX) markers in mouse brains was quantified via immunohistochemical analyses with the specific antibodies.

Alteration in specific brain regions after prenatal treatments were found via immunohistochemistry and were similar to those observed in cultured neurons. Indeed, comparing the synaptophysin levels between the control and mice exposed to the pesticide, imidacloprid did not affect synaptophysin expression in the hippocampus and cerebellum at the lowest tested dose (H. exp., 118 µg/kg bw/inj), while at ADI (4.1 mg/kg bw/inj), the expression of the signal intensity of the presynaptic marker was increased by about 60% and 80%, respectively (*p* < 0.001 compared to the control group; Figure 10A,B). Treatment with 10X ADI induced an increase in synaptophysin levels only in the cerebellum (*p* < 0.05 vs. CTRL). Nicotine induced an increase in synaptophysin expression at the lower dose (26.5 µg/kg bw/inj; *p* < 0.05 vs. CTRL) only in the hippocampus and at the higher dose (53 µg/kg bw/inj) in both brain regions (*p* < 0.001 and *p* < 0.05 vs. CTRL, respectively).

Impairments in neurogenesis markers were also induced in the hippocampal region starting at the same experimental dose (ADI), as assessed by decrease in the area stained with DCX (*p* < 0.05 versus CTRL; Figure 11A,B). Nicotine reduced DCX expression only in the hippocampus at the higher dose tested (*p* < 0.05 vs. CTRL). No significant effect was revealed in the cerebellum.

Specific microglia and astrocyte alterations were also evaluated in different brain regions (i.e., the hippocampus and cerebellum) by immunohistochemistry for IBA-1 or GFAP. Considering the effects on microglia, imidacloprid decreased the IBA-1-positive cell density at 4.1 mg/kg bw/inj (ADI) and 10× ADI in the hippocampus (*p* < 0.01 and *p* < 0.05 versus CTRL, respectively) and cerebellum (*p* < 0.05 versus CTRL, both doses) (Figure 12A,B). A significant reduction was also induced by the highest dose of nicotine (53 µg/kg bodyweight/inj) both in the hippocampus and cerebellum (*p* < 0.001 and *p* < 0.01 versus CTRL, respectively; Figure 12A,B).

## 4. Discussion

In this project, different primary cultures of CNS cells and a mouse model of prenatal exposure were used to investigate the effects induced by early exposure to neonicotinoids on neuron development. Preliminary evidence of imidacloprid neurotoxicity on human iPSC-derived neural cultures and brain organoids was also provided. Since neurodevelopment is a highly complex biological process, a single type of endpoint and one single cell or tissue type are unlikely to be sufficient for a comprehensive testing of neurodevelopmental toxicity. Thus, we investigated here the neonicotinoid-mediated neurotoxicity at distinct levels of analysis via complementary techniques. To provide direct relevance for human exposure to the pesticides, the determination of the experimental doses of neonicotinoids to be tested was based on the estimated levels in the diet of women of childbearing age and were compared here with increasing doses (up to 10,000 times the admissible intake) to detect the impact of a real-life neonicotinoid exposure.

Several studies have shown a widespread expression of nAChRs in CNS tissues and cells, their specific and complex organization, and their relevance to normal brain development. Here we confirmed that nAChRs are expressed in different growing neuron populations as well as in microglia. Interestingly, astrocytes of human origin do express the receptors, while those form mouse tissues do not, thus supporting a possible species-specific toxic impact on the development of these cells.

Although neonicotinoids have been developed to selectively interfere with nAChR signaling in insects, some recent evidence suggests that neonicotinoids may affect mammalian nAChRs to a greater extent than previously believed (based only on binding-assay data) and represent a potential risk for neurodevelopment in humans [1,34,60]. In this sense, in-depth toxicological studies on neonicotinoids would greatly improve the knowledge of their effects on key developmental processes and determine the putative neurotoxic pathways and modes of actions involved. We showed here that neonicotinoids are toxic both on early- and late-developing neurons. At the latter developmental stage, neonicotinoids are able to induce a significant reduction in neuron viability by lower concentrations than in early-developing neurons. Interestingly, imidacloprid and dinotefuran induced significant neurotoxicity down by the estimated (eFBC, 5 nM) or admissible human fetal levels (aFBC, 170 nM or 620 nM for imidacloprid or dinotefuran, respectively) in hippocampal and cortical cultures. These data could be explained by a different vulnerability of the tested fetal brain areas (the hippocampus, cerebellum, and cortex), as well as by a different degree of expression and functionality of the nAChRs during neuronal maturation. The direct involvement of nAChRs in neonicotinoid-mediated neurotoxicity was verified via co-treatments with the specific non-competitive antagonist mecamylamine that was able, in our experimental model, to completely reverse the neurotoxic effects induced by imidacloprid or nicotine in hippocampal neurons.

The “sandwich” coculture models, established by combining primary neurons suspended above a feeder layer of glial cells, allowed us to obtain low-density neurons suitable to investigate possible neurodevelopmental alterations induced by low concentrations of the neonicotinoids on dendritic arborization and synaptic connections. The preliminary results showed that a micromolar concentration of imidacloprid (17 µM) might also induce non-lethal neuronal damage, such as impairments in synaptic formation. In fact, it increased the expression levels of presynaptic proteins (synaptophysin) both as volume and as number of puncta compared to the control condition.

Studies published so far have showed neurotoxic effects in micromolar levels. Consistently, concentrations of neonicotinoids ranging from 1 to 10 μM exert excitatory effects in mammals by targeting nAChRs [25,26,27,28]. In our experimental model, neonicotinoids induced toxic effects on neuronal mammalian cells from nanomolar concentrations, suggesting that early exposure to low levels of neonicotinoids during the critical phases of neurodevelopment might be responsible for CNS region-specific impairments in the maturation mechanisms of neurons and correct synaptic assembly. We also showed for the first time that low levels of imidacloprid might affect the development of human iPSC-derived neurons and astrocytes in cultures. As the dose of exposure increases, the neural cell viability is also impaired in complex 3D tissue-like structures like the brain organoids, thus posing evidence of human CNS cell susceptibility to neonicotinoids.

We further assessed whether, in addition to the neurodevelopmental alterations, neonicotinoids can affect glial viability and induce alterations on the activation status of CNS immune competent cells by evaluating the mRNA expression of pro-/anti-inflammatory markers and the morphological parameters that characterize the different phenotypes of microglia. Interestingly, microglia were highly sensitive to neonicotinoids, showing a significant reduction in cell viability down by eFBC for imidacloprid (5 nM) and aFBC for dinotefuran (620 nM), while exposure to neonicotinoids (up to 1.7–6.2 mM) did not affect astrocyte survival. Moreover, although imidacloprid did not directly affect the microglial phenotype, it may induce immune-depression effects decreasing the pro-inflammatory responses of immune cells to LPS following pre-incubation at the higher dose tested (170 µM). These data are consistent with the immunomodulatory effects already reported in the literature for nicotine and other weak agonists of nAChRs [66,67] and strongly suggest a multi-target toxicity by this class of insecticides that also involve immune cells. Consequently, neonicotinoids may lead to immune dysregulation and increase host susceptibility to diseases and immune disorders by doses similar to those that women of childbearing age could be exposed to.

Although to different extent, the results on neurodevelopmental alterations observed in cultured CNS cells were confirmed in vivo. Here, the levels of expression of pre-/postsynaptic and neurogenesis proteins were assessed via immunohistochemistry on single brain samples isolated from prenatally exposed newborn mice. Comparing the expression of synaptic proteins between the control and treated embryos, imidacloprid increased the expression of the presynaptic marker synaptophysin with ADI. Since this synaptic marker is involved in synapse formation and synaptic plasticity, imbalances in its level even after the exposure to low imidacloprid doses suggest that it represents a concerning environmental risk that might also affect the long-term development of the brain and result in functional impairments later in life. The same experimental doses that affected the expression of synaptic proteins (ADI) induced dysregulation in a neurogenesis marker, as evidenced by the reduction in doublecortin levels. These data, together with the increase in the levels of synaptic proteins, suggest that imidacloprid may drive impairment in neuron maturation. Moreover, evidence from the in vivo model of fetal exposure through maternal treatment further strengthens the hypothesis that imidacloprid may affect immunocompetent cells at low levels of exposure, as highlighted by the reduction in IBA-1-positive cells in the hippocampus and cerebellum starting from ADI. Interestingly, prenatal exposure to nicotine showed comparable effects at doses that could be relevant for smokers.

Thus, these data showed that the exposure to low levels of neonicotinoids (representative of human fetal exposure or the maximum admissible daily intake) might be responsible for impairments during the critical phases of neurodevelopment/immune development, supporting the reported observation of a positive association between mothers’ imidacloprid exposure during pregnancy and adverse birth outcomes [68,69,70]. Taken together with our recent report of the ability of imidacloprid to pass the biological barriers and distribute from maternal blood to fetal tissues [43], the results reported here are of relevance for the characterization of a potential toxic impact of neonicotinoid exposure on neuron development and provide preliminary evidence of risk for imidacloprid.

## Data Availability

The datasets generated and/or analyzed during the current study are available in Zenodo repository (https://zenodo.org/records/12635655 accessed on 1 January 2024).

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
