# Peer review of "Neonicotinoid Pesticides Affect Developing Neurons in Experimental Mouse Models and in Human Induced Pluripotent Stem Cell (iPSC)-Derived Neural Cultures and Organoids"

_cells, 2024, doi:10.3390/cells13151295_

Round 1

Reviewer 1 Report

Comments and Suggestions for Authors

This article focuses on the developmental toxicity of Neonicotinoid pesticide, clearly shown the neuron viability, neurofilament structure, synaptic protein expression ad microglial activation (In vitro and in exposed mouse fetuses). 

Abstract, introduction, methodology, result and discussion were well written and easy to follow. I think the introduction is bit lengthier; I would appreciate if the introduction were little, short. Also, by including recent statistics or references regarding the prevalence of neurodevelopmental disorders and their possible link to environmental contaminants. Breaking long sentences would make it much easier to follow. 

Methods is detailed and clear. Ensure including the company name and catalog number of the chemicals and antibodies which are missing for some chemicals. Ensure all abbreviations are defined upon their first use (e.g., CCM for complete culture medium).

Mentioning the region of hippocampus (DG, CA1, CA2 or CA3) and the region of cortex and cerebellum would be recommended. Kindly mention the number of replicates (n) in the legend.

Discussion is well written except for the long sentences at some points, else it is clear and understandable.

Most part of the article is well written except the above-mentioned minor corrections. 

Comments on the Quality of English Language

Well written with appropriate use of words

Reviewer 2 Report

Comments and Suggestions for Authors

In the manuscript, authors investigated the neurodevelopmental effects of different generations of neonicotinoids on CNS cells in mouse fetal brain and primary cultures, and in neuronal cells and organoids obtained from human induce pluripotent stem cells (iPSC). The results show that the exposure to low levels of neonicotinoids might be responsible for impairments during the critical phases of neuro/immune development in mouse and human CNS cells. This is a very interesting and meaningful study, and it provide new insights in the characterization of risk for the exposure to this class of pesticides. However, some issues should be addressed prior to publication.

Major revision:

1. The method section 2.8 Immunohistochemistry, lacks sufficient detail, making it less descriptive. I suggest providing more comprehensive details to enhance the reproducibility of the study.

2. P6 Line 251-252, about Nicotine, Low Dose and High Dose, the author should explain how this measurement was determined.

3. P6 Line 260-262, Serial paraffinized 20 µm-thick sagittal brain slices were collected, and freely floating brain slices were stained. It is difficult or almost impossible to stain paraffin sections using the freely floating method, here the description of the method needs to be clarified.

4.Results 3.3 Imidacloprid induces neurodevelopmental alterations in mouse neuron cultures. Figure 6. Cells were stained with DCX, NF200 and synaptophysin antibodies; nuclei were counterstained with Hoechst 33258 dye. It is necessary to provide representative images of the above results.

5. Figure 10 (A), Figure 12 (A), what is the staining method used here?  It is not described in the method section.

Minor revision:

1. P4 Line 185, S100B; P12 line 466, S100b; S100b in Figure 7. A, P13 line 470, S100b-positive i-astro----etc. Pay attention to the spelling for S100β.

2. Figure 10 (A) shows the cerebellum bellow, but it doesn't look like a image of the cerebellum, it's more like a picture of the cerebral cortex. So the author needs to confirm or replace this picture.

Reviewer 3 Report

Comments and Suggestions for Authors

Ref: cells-3115611

Title: Neonicotinoid pesticides affect developing neurons in experimental mouse models and in human induced pluripotent stem cell (iPSC)-derived neural cultures and organoids

Studies have shown that high exposure levels during fetal development can lead to neurotoxicity in animal models, necessitating further investigation into their impact on the mammalian central nervous system. Research on mouse cell cultures reveals that neonicotinoids, particularly imidacloprid (IMI), significantly affect neuron viability. IMI induces alterations in synaptic protein expression, disrupts neurofilament structures, and activates microglia in vitro and in prenatally exposed mouse fetuses. Similar neurotoxic effects were observed in developing human neurons and cortical organoids derived from induced pluripotent stem cells (iPSC), suggesting a risk to human neurodevelopment.

What deserves special attention is the very well-thought-out calculation of the dose of neonicotinoid used for subsequent tests (Table 1)

In my opinion, the paper is very good and will attract many readers. Nevertheless, to enhance its quality further, I would recommend some corrections, as provided below.

Comments:

1.      Section 2.3 is written in a very general manner. Additionally, it is unnecessary to place important information, such as the ingredients of the cell medium, in the supplementary section.

2.      Please provide complete and specific protocols to ensure that the results obtained can be replicated by another scientific group.

3.      Adding some schemes would be appreciated, as the experimental model is quite complex.

4.      Regarding the qPCR, please provide more information on the chemicals used, the method of selecting the reference gene, etc.

5.      Please add the aim of the study as a separate part of introduction.

6.      Given that the authors used embryos at an early stage of development (12 ED), what are the chances that the cells at 6 and 12 DIV are still pre-neurons, pre-glia, etc., and express nearly all markers?
